# A Predictive Model of Adaptive Resistance to BRAF/MEK Inhibitors in Melanoma

**DOI:** 10.3390/ijms24098407

**Published:** 2023-05-07

**Authors:** Emmanuelle M. Ruiz, Solomon A. Alhassan, Youssef Errami, Zakaria Y. Abd Elmageed, Jennifer S. Fang, Guangdi Wang, Margaret A. Brooks, Joe A. Abi-Rached, Emad Kandil, Mourad Zerfaoui

**Affiliations:** 1Department of Pathobiological Sciences, School of Veterinary Medicine, Louisiana State University, Baton Rouge, LA 70803, USA; 2Department of Surgery, Tulane University School of Medicine, New Orleans, LA 70112, USA; 3Department of Pharmacology, Edward Via College of Osteopathic Medicine, University of Louisiana, Monroe, LA 71203, USA; 4Department of Cell and Molecular Biology, Tulane University School of Science & Engineering, New Orleans, LA 70118, USA; 5Department of Chemistry, RCMI Cancer Research Center, Xavier University of Louisiana, New Orleans, LA 70125, USA

**Keywords:** melanoma, BRAF/MEK inhibitors, aggressiveness, risk stratification, resistance, biomarkers

## Abstract

The adaptive acquisition of resistance to BRAF and MEK inhibitor-based therapy is a common feature of melanoma cells and contributes to poor patient treatment outcomes. Leveraging insights from a proteomic study and publicly available transcriptomic data, we evaluated the predictive capacity of a gene panel corresponding to proteins differentially abundant between treatment-sensitive and treatment-resistant cell lines, deciphering predictors of treatment resistance and potential resistance mechanisms to BRAF/MEK inhibitor therapy in patient biopsy samples. From our analysis, a 13-gene signature panel, in both test and validation datasets, could identify treatment-resistant or progressed melanoma cases with an accuracy and sensitivity of over 70%. The dysregulation of HMOX1, ICAM, MMP2, and SPARC defined a BRAF/MEK treatment-resistant landscape, with resistant cases showing a >2-fold risk of expression of these genes. Furthermore, we utilized a combination of functional enrichment- and gene expression-derived scores to model and identify pathways, such as HMOX1-mediated mitochondrial stress response, as potential key drivers of the emergence of a BRAF/MEK inhibitor-resistant state in melanoma cells. Overall, our results highlight the utility of these genes in predicting treatment outcomes and the underlying mechanisms that can be targeted to reduce the development of resistance to BRAF/MEK targeted therapy.

## 1. Introduction

An estimated 325,000 new cases and 57,000 deaths from melanoma were recorded worldwide in 2020, with significant regional and racial disparities in incidence and mortality rates [1]. Although the prognosis and survival rate for this cancer type is high (an overall 5-year survival rate of 93% for localized melanoma), primary and acquired resistance to treatment is a major challenge in clinical care and significantly contributes to poorer outcomes, particularly at the metastatic stage. Compared to the early-stage of cutaneous melanoma, where tumors are often accessible and can be resected, treatment options for metastatic-stage melanomas are limited and previously relied on drugs, such as the monofunctional alkylating agents temozolomide and dacarbazine, with often poor results [2]. The introduction of targeted inhibitors for the mutated oncogenic proteins that drive melanoma onset and aggressiveness, such as BRAF (vemurafenib and dabrafenib), MEK (trametinib and cobimetinib), and c-KIT (imatinib), has improved the durability of treatment response [3]. Additional classes of therapies, such as immune checkpoint inhibitors and the use of oncolytic viruses [4], also improved clinical response [5,6]. Although successful targeting of BRAF^V600E^ and MAPK pathway improves overall survival [7,8,9], de novo resistance (intrinsic or primary) occurs in up to 20% of patients, and acquired resistance (secondary) is shown to rapidly emerge in the vast majority of patients [10,11,12,13,14,15]. The acquisition of resistance is driven by multiple mechanisms. These mechanisms include the over-expression of BRAF^V600E^ or utilization of its splice variants [16], activation of Hgf/Met signaling pathways [17], and activating or gain-of-function mutations in *NRAS*, *KRAS*, *MEK1*, and *AKT1/AKT3* [18]. Other mechanisms include overexpression of receptor tyrosine kinases and melanocyte transcription factor (MITF) [19], as well as epigenetic reprogramming [20]. Furthermore, tumor heterogeneity and transcriptomic and metabolic plasticity may accelerate adaptive resistance to BRAF and MEK inhibitors, though these events are poorly characterized [21].

Targeted therapy reshapes transcriptional programs in cancer cells and the surrounding tumor environment, underpinning tumor phenotypes arising from drug sensitivity or resistance. High throughput gene quantification methods, such as microarrays and RNAseq, allow for unbiased interrogation of the transcriptomic landscape of cancer cells/tissues exposed to targeted therapy [19]. Identifying markers that predict the progression of melanoma resistance stage during targeted therapy are urgently needed for making informed decisions on treatment strategy, patient management, and disease monitoring protocols.

In our previous study [22], we identified 18 differentially abundant proteins between PLX-4032 (vemurafenib)-sensitive and PLX-4032-resistant A375 melanoma cell lines. In this current study, we evaluated the expression, predictive potential, and functional roles of genes corresponding to this panel of proteins, using publicly available transcriptomic datasets from tissue biopsy samples of patients treated with BRAF or MEK inhibitors. We hypothesize that significant and clinically relevant differences exist in the expression patterns of these genes, which are suggestive of a mechanistic program through which these gene signatures mediate treatment resistance and progression in melanoma.

## 2. Results

### 2.1. The 18 Proteomic Panel Is Deregulated in the Resistant Human Melanoma Model

The 18 genes selected from the proteomic analysis on A375 cell lines (Table 1) were analyzed in the patient-derived datasets. Kruskal–Wallis tests and linear regression analyses were performed to compare their expression before treatment (PRE) and after relapse (PROG). Five genes had significant (*p*-value < 0.1) odd ratios (OR) in progressed biopsies compared to pre-treatment samples: *HMOX1* (OR = 2.68, *p* = 7.4 × 10^−2^), *ICAM1* (OR = 1.37, *p* = 3.6 × 10^−2^), *MMP2* (OR = 1.12, *p* = 2.9 × 10^−2^), *TP53* (OR = 1.09, *p* = 7 × 10^−2^), and *SPARC* (OR = 0.37, *p* = 2.5 × 10^−2^) (Figure 1A). The Kruskal–Wallis test confirmed linear regression analyses with a significant difference between PRE and PROG samples (*HMOX1*, *p* = 5.7 × 10^−2^; *ICAM1*, *p* = 6.90 × 10^−2^; *MMP2*, *p* = 6 × 10^−3^; *TP53*, *p* = 9.0 × 10^−2^; *SPARC*, *p* = 1.6 × 10^−2^). *HMOX1*, *ICAM1*, *MMP2*, and *TP53* were upregulated in PROG samples, whereas *SPARC* was downregulated (Figure 1B).

A PCA was performed on the 30 samples using the 18 genes. The two first principal components (PC1 and PC2) explained 40.4% of the variance observed across the samples (Figure 1C). Interestingly, PC1 significantly differentiated the samples according to the treatment stage (PROG vs. PRE) (*p*-value = 7.9 × 10^−3^). Eight genes actively contributed to the variance observed as their contribution was higher than average (Figure 1D). *HMOX1*, *ENG*, *ICAM1*, and *GRN* (purple ellipse) formed a subgroup of genes separate from *MMP2* and *TP53* (green ellipse). Both gene types have an inverse correlation with *VIM* and *SPARC* (orange ellipse). This observation was confirmed via Spearman’s correlation analysis and hierarchical clustering (Figure 1E). *HMOX1*, *ENG*, *ICAM1*, and *GRN* were strongly correlated and formed one cluster (cluster 1 in purple square). The significant Spearman’s coefficients were 0.69 for *GRN* and *ICAM1*, 0.60 for *GRN* and *ICAM1*, 0.60 for *ICAM1* and *ENG*, 0.52 for *ICAM1* and *HMOX1*, and 0.80 for *ENG* and *HMOX1*. *VIM* and *SPARC* had a significant correlation (rho = 0.36) and were clustered together (cluster 3 in the orange square). Interestingly, cluster 1 was positively correlated with cluster 2 (in green square), with significant correlations between *MMP2* and *GRN* (rho = 0.47), *MMP2* and *ICAM1* (rho = 0.57), *MMP2* and *ENG* (rho = 0.68), *MMP2* and *HMOX1* (rho = 0.69), and *TP53* and *GRN* (rho = 0.38). Finally, clusters 1 and 2 were inversely correlated with cluster 3, with significant correlations between *MMP2* and *VIM* (rho = −0.76), *TP53* and *SPARC* (rho = −0.39), *TP53* and *VIM* (rho = −0.38), *ENG* and *VIM* (rho = −0.56), *HMOX1* and *VIM* (rho = −0.59), and *ICAM1* and *SPARC* (rho = −0.56). These results suggest that therapeutic resistance is accompanied via increased expression of cluster 1 and 2 genes and decreased expression of cluster 3 genes. This observation is strengthened via the distribution of the 18 genes’ expression between PRE and PROG samples, where genes for clusters 1 and 2 are upregulated, whereas genes from clusters 3 are downregulated in PROG samples compared to PRE samples (Figure 1B).

Classifier predictions were performed with a linear discriminant analysis (LDA) regression, followed by the estimation of optimized classifier markers—the AUC of a ROC analysis, the sensitivity, the specificity, the PPV, and the NPV of the 18 genes (Table 2). The eight genes driving the variation observed with the PCA had an AUC ranging from 0.624 to 0.796, a PPV ranging from 0.6 to 0.78, and an NVP ranging from 0.68 to 0.82.

### 2.2. The 13-Risk Panel Predicts the Resistance Stage

From the 18 genes corresponding to proteins identified in our proteomic data, we identified a combination of 13, now named 13-risk signature (Figure 2A), that differentiated the samples with an AUC of 0.982 (95%CI = 0.943–1.021). The signature displayed a sensitivity of 0.92, specificity of 1, PPV of 1, and an NPV of 0.94 in the primary dataset (Table 3—first row). The 13-risk signature score was higher in PROG samples with an adjusted *p*-value of 4.26 × 10^−5^ (Figure 2B). Four other transcriptomic datasets, containing samples from patients before and after acquiring resistance to BRAF inhibitors, were analyzed to validate the 13-risk signature (Figure 2C). This signature was found to be upregulated in PROG samples with adjusted *p*-values of 7.0 × 10^−3^, 2.53 × 10^−4^, 1.0 × 10^–3^, and 2.1 × 10^−2^ for GSE61992, GSE50509, GSE65185, and GSE77940, respectively. In the “FiveDatabase” using z-score values, the 13-risk signature was highly overexpressed in PROG samples compared to PRE samples (adjusted *p*-value = 3.76 × 10^−6^) (Table 3). A linear regression was also performed to verify the overexpression of this signature in the drug-resistant stage. In the original dataset (GSE99898), the 13-risk signature had an OR of 7.38 (95% CI = 3.58–15.19; adjusted *p*-value = 2.10 × 10^−5^), and was highly expressed in PROG samples and significantly dysregulated in the other data sets. In the merged databases, the 13-risk signature had a 2-fold probability of being highly expressed in relapsed/resistant samples (OR = 2.11, CI-95% = 1.56–2.86; adjusted *p*-value 8.63 × 10^−6^) (Table 3—Figure 2D). Finally, classifier markers were estimated. The 13-risk signature had an AUC of 0.71 in the combined dataset, with a PPV of 73% and an NPV of approximately 68%, corresponding to strong values (Table 3—Figure 2E). These analyses’ observations suggest a strong association between the 13-risk signature and the occurrence of resistance to BRAF/MEK targeted treatment in melanoma patients.

### 2.3. The 13-Risk Score Is Highly Differentiated according to the Resistance Stage

In the GSE99898 dataset, the expression level of the 13-risk signature explained approximately 40% of the variance (*p* = 7.87 × 10^−3^), was observed with a PCA, and differentiated the samples according to the treatment stage (Figure 3A). The differentiation of the PROG samples was predominantly driven by eight genes that have a strong active contribution to the PCA results (Figure 3B). *HMOX1*, *MMP2*, *ENO2*, *GRN*, and *TP53* were highly associated with PC1, meaning that their overexpression was associated with the resistance rate. However, *SPARC*, *DKK1*, and *CXCL8* were inversely associated with PC1, meaning their overexpression was associated with samples before treatment. The Spearman’s correlation matrix supported these observations. Indeed, *HMOX1*, *MMP2*, *ENO2*, *GRN*, and *TP53* were in the same cluster and were positively correlated, with significant Spearman’s coefficients for *HMOX1* and *MMP2* (rho = 0.69), *GRN* and *MMP2* (rho = 0.47), *GRN* and *TP53* (rho = 0.38), and *GRN* and *ENO2* (rho = 0.46). On the other hand, *CXCL8*, *SPARC*, and *DKK1* were clustered separately, with significant Spearman’s coefficients for *DKK1* and *CXCL8* (rho = 0.43) (Figure 3C). The potential of the signature to differentiate samples according to the resistance stage, through PCA analysis, was also confirmed in the “FiveDatabase” (Figure 3D–F). In this case, the signature explained 29% of the variance observed between the samples. Interestingly, the 13-risk signature significantly separated the samples according to the treatment stage (*p*-value = 1.13 × 10^−3^). As shown in Figure 3G,H, some of the 13 genes in the panel are differentially expressed according to this specific risk score FC pattern. In the PRE samples, *HMOX1* was significantly more expressed in the samples with a “UP” 13-risk score FC (*p* = 4.0 × 10^−2^). The “FiveDatabase” showed similar results, with a highlight of CLXL8 (*p*-value = 4.5 × 10^−2^), *EGFR* (*p* = 3.9 × 10^−2^), *ENO2* (*p*-value = 2 × 10^−3^), and *TP53* (*p*-value = 1.2 × 10^−2^) differentiation. The 13-risk score FC-mediated PROG samples stratification could be characterized as a differential expression of *BIRC5* (*p* = 3.1 × 10^−2^), *GRN* (*p* = 8.0 × 10^−3^), *HMOX1* (*p* = 2.0 × 10^−2^), *MMP2* (*p* = 2.8 × 10^−4^), *SPARC* (*p* = 1.0 × 10^−3^), and *TP53* (*p* = 7.0 × 10^−3^).

### 2.4. Signaling Pathways and Therapeutic Sensitivity

Active-subnetwork-oriented pathway enrichment analysis was performed on the differentially expressed genes obtained from the GSE99898. The analyses were performed independently of the 13-risk signature. However, to understand which transcriptomic landscape is associated with the 13 gene panel, only the pathways in which an enrichment score significantly correlated with the 13-risk gene signature were selected. The same analysis was performed with the differentially expressed genes in the combined dataset (“FiveDatabase”). A total of 19 pathways were identified as associated with BRAF inhibitor resistance mediated by using the 13-risk gene signature. The top 12 pathways are shown in Figure 4A. After completing a PCA, we observed that the scores for these 12 pathways explained more than 80% of the variance observed across the samples, and significantly differentiated them according to their resistance status (*p* = 3.0 × 10^−3^) (Figure 4B). A hierarchical clustering was performed on the PCA, and three distinct clusters were identified, all significantly associated with the resistance stage (Figure 4C). Cluster 1 was strongly enriched in PROG samples and characterized by the enrichment terms “Proteasome”, “Alzheimer disease”, and “Parkinson disease” pathways, and the impoverishment of “Measles”, “Toxoplasmosis”, “Circadian rhythm”, “Focal adhesion”, “Proteoglycans in cancer”, “Necroptosis”, “Hippo signaling pathway”, and “Toll-like receptor signaling pathway” entries. In contrast, cluster 3, which was strongly enriched in PRE samples, presented an opposite pathway signature to that of cluster 1. Finally, cluster 2 was characterized by the “Fc-gamma receptor-mediated phagocytosis” (Figure 4C). The 13-risk signature score was significantly associated with the pathway PCA clusters (*p* = 9.4 × 10^−4^), as well as the pathway PCA index score (rho = 0.76, *p* = 3.57 × 10^−6^), with a higher score found in PROG-enriched cluster 1 (Figure 4D).

As previously shown in Figure 2E, the 13-risk score genes had a specific pattern of fold change (downregulated, slightly upregulated, or upregulated) in PROG samples when compared to PRE samples. (Figure 4E). Plotting each sample based on its risk score and PCA scores derived from pathway enrichment, we observe that the pathway PCA score was significantly inversely correlated with the 13-risk score FC at the progressed/drug-resistant stage (rho = −0.58, *p* = 3.8 × 10^−2^) (Figure 4E—right). These results were confirmed when the merged datasets were analyzed (Appendix A).

### 2.5. Pathways Score Distribution according to the 13-Risk Score

To understand which molecular mechanisms explain the difference between samples sensitive or resistant to BRAF/MEK inhibitors, we carried out functional enrichment analysis to identify specific pathways that were associated with the 13-risk signature score. Toward this aim, Kruskal–Wallis rank sum tests were performed with the pathway scores in the PROG samples according to the fold change strata, i.e., downregulated, slightly upregulated, and upregulated. Among the twelve pathways commonly correlated with the 13-risk score in the GSE99898 and the “FiveDatabase”, six presented a significantly lower score between the PROG samples. These include T “Focal adhesion” (*p* = 5.0 × 10^−3^), “Proteoglycan in cancer” (*p* = 1.9 × 10^−2^), “Hippo signaling pathway” (*p* = 1.9 × 10^−2^), “Toll-like receptor signaling pathway” (*p* = 2.8 × 10^−2^), “Circadian rhythm” (*p*-value = 2.8 × 10^−2^), and “Measles” (*p* = 2.8 × 10^−2^) (Figure 5). Similar significances were found in the “FiveDatabase” analysis (Appendix A). Furthermore, other pathways presented significant differences according to the fold change strata. These pathways include “Cell cycle” (*p*-value = 1.3 × 10^−2^), “Estrogen receptor signaling” (*p*-value = 8.0 × 10^−3^), “mRNA surveillance pathway” (*p*-value = 2.3 × 10^−2^), “Ubiquitin mediated proteolysis” (*p*-value = 1.9 × 10^−2^), “Longevity regulating pathway—multiple species” (*p*-value = 1.9 × 10^−2^), “Chronic myeloid leukemia” (*p*-value = 8.0 × 10^−3^), and “Yersinia infection” (*p*-value = 1.3 × 10^−2^) (Figure 5, Appendix A).

### 2.6. The 105. Genes Involved in the Resistance-Related Pathways

The pathways previously selected were strongly correlated with the 13-risk gene panel, as shown in Figure 6A, highlighting two sub-clusters of genes. The first sub-cluster of genes was positively associated with the enriched pathways and included the genes GRN, HMOX1, MMP2, TP52, and BIRC5. The second sub-cluster included the genes SPARC, CXLC8, and CAPG. To decipher the gene network surrounding the 13-gene panel and understand their role in melanoma therapeutic resistance, genes involved in the 19 pathways presenting a significant correlation with the 13-highly predictive genes and significant expression differences in PRE and PROG samples (in all datasets) were collected. A total of 105 genes matched these criteria. We observed two clusters of genes presenting a different expression pattern according to the sample resistant stage and pathway PCA clusters. Interestingly, the two gene clusters were strongly differentially expressed in the pathway PCA cluster 1 (Figure 6B). Example of genes upregulated in PROG samples in this interaction network included *BIRC5* and *BAK1* (negative regulators of mitochondrial apoptosis), *HMOX1* and its regulator *PGAM5*, and *MMP2*.

### 2.7. A Novel Model of Therapeutic Resistance

Combining the PPI network and pathway analysis, a model of therapeutic resistance, dependent on the 13-risk panel, was proposed (Figure 7). Although tumors progressing despite treatment are largely defined based on the acquisition of additional de novo gene mutations, which cause the reactivation of the MAPK signaling or activation of other survival pathways, transcriptomic alterations and signaling pathway modifications are also integral to BRAF or MEK inhibitor resistance.

Our model suggests a possible role of mitochondrial dysfunction/stress and the rewiring of mitochondrial energy hemostasis (a well-known hallmark of cancer cells) [23] in the emergence of BRAF inhibitor resistance in melanoma. We propose a strong role for HMOX1 in metabolic plasticity and the ability of melanoma cells to switch metabolic phenotypes, as these cells adapt or initiate mechanisms to resist BRAF/MEK inhibitor treatment. This proposal is plausible due to the known functional roles of HMOX1 and the activity of its upstream regulators PGAM5 and NFEL2L2 (NFR2). Firstly, HMOX1 is induced through diverse cellular stress, such as drug pressure, and mediates anti-inflammatory antioxidant (through eliminating cellular heme), and anti-apoptotic processes in cells, enhancing cell survival and adaptation [24]. Furthermore, HMOX1, via NFR2 (NFE2L2) and Akt, was demonstrated to stimulate mitochondrial biogenesis and mitophagy, thereby modulating the pool of healthy mitochondria in cells [25,26]. Evidence also suggests that the controlled ability of cells to clear out damaged mitochondria (via induction of HMOX1 and other mitophagic mechanisms) contributes to adaptive drug resistance [24,27].

Regulation of HMOX1 abundance and activity is proposed to occur via modulation of NFE2L2, which is a transcription factor that plays a key role in the response of oxidative stress and mitochondrial function and dynamics. However, NFE2L2 protein stability and function are highly regulated through its degradation process. The main mechanism is mediated via KEAP-1 E3 ligase. However, another E3 ligase—β-TrCP-CUL1-based ubiquitin E3 ligase—was also found to target NFE2L2 [28,29]. Thus, our model proposes that the regulation of HMOX1 abundance via NFE2L2 may contribute to adaptive resistance under chemotherapeutic pressure.

Our model also postulates that the involvement of matrix metalloproteinases and EGFR signaling can contribute to the development of BRAF/MEK inhibitor resistance in melanomas. Studies have shown that MMP-2 overexpression in progressed/resistant stage samples activates EGFR signaling and downstream pathways, including MAPK, MEK/ERK and PI3K/AKT, which are involved in BRAF/MEK inhibitor response. The downstream consequences of these activations include the transcription of anti-apoptotic and pro-proliferation genes, including survivin/BIRC5 (which is also dysregulated in progressed/resistant samples). Indeed, the inhibition or activation of MEK was demonstrated to regulate the expression of regulators of cell proliferation and apoptosis, such as BIRC5 [30]. This finding is plausible given that EGFR-dependent signaling activations are implicated in resistance to several tyrosine kinase inhibitors [31]. The gene *TP53*, which is a member of the 13-risk gene panel, presented a higher expression in resistant samples and plays the role of the cell cycle guardian. As ATM, the principal activator of p53, is downregulated, p53 overexpression could be the result of the mutations. However, in the context of this project, it was not possible to analyze the mutation status of p53. Interestingly, the overexpression of the Bcl-2 homologous antagonist/killer (BAK) was observed in resistant samples. BAK plays a role in mitochondrial apoptosis through promoting the mitochondrial outer membrane permeabilization, realizing apoptotic factors as cytochrome C [32].

## 3. Discussion

We employed a hypothesis-driven approach to interrogate the gene expression and predictive potential of a panel of 18 cancer-associated proteins previously detected in a BRAF inhibitor-resistant melanoma cell line [22]. This study was carried out in patient-derived datasets through comparing the level of this panel signature before treatment and after the emergence of resistance to BRAF/MEK inhibitors. Firstly, we highlighted important differences in the transcriptomic landscape initiated via treatment and consequent rewiring/reorganization of the transcriptional pattern during drug resistance and recurrence. For example, HMOX1 and SPARC were identified as the most up- and down-regulated genes, respectively, dysregulated during the progression from BRAF/MEK inhibitor sensitivity to resistance/progression. The cells’ addiction to oncogene-driven, genome-wide transcriptional changes was observed as a feature of melanomas (and most other cancer types) during the stages of initiation, progression, metastasis, and therapy resistance [33]. Effective transcriptional signaling and transcriptional plasticity ensure that melanomas can quickly adapt gene expression patterns to deal with pressures, such as when key driver kinases are targeted during therapy [34]. Through leveraging gene signatures in predicting the durability of drug response, this study also identified the expression levels of the genes MMP2, SPARC, and HMOX1 as having a good predictive value in predicting the emergence of chemoresistance to BRAF/MEK inhibitor therapy in melanomas. This finding could be linked to the critical role of these genes and their interactors in mechanisms that drive adaptive drug response in cancer cells.

Several mechanisms for chemotherapy resistance with the critical involvement of HMOX1 were elucidated or suggested. We previously showed that mutant BRAF drives the overexpression of HMOX1, which, in turn, forces the nuclear localization of mutant BRAF and drives proliferation and metastasis in melanomas via reactivation or persistent activation of MAPK signaling [22]. Other researchers have suggested that JUND-mediated transactivation of HMOX1 is an adaptive mechanism that regulates the resistance to cisplatin-based chemotherapy in muscle-invasive bladder cancer [35]. This mechanism might also be present in melanomas. An additional role for HMOX1 in the emergence of drug resistance is in the dysregulation of mitophagy and mitochondrial energy homeostasis. Cells can adapt to selective pressure and mediate therapy resistance through optimizing mitophagy to efficiently break down and recycle damaged mitochondria and mitochondrial components. [26]. Furthermore, proteomic studies show HMOX1 interacts with the HIF1A-BNIP3-ATG7 axes to control mitophagic flux and anti-apoptotic effects [36]. Another consequence of HMOX1 dysregulation with an effect on the sensitivity of tumor cells to therapy is the modulation of drug transport. For example, resistance to sorafenib (a protein kinase inhibitor) in hepatocellular carcinoma was demonstrated to be mediated through HMOX1 regulation of ABC transporters in HCC cells [37].

An additional mechanism in the emergence of resistance involves changes in the expression of SPARC. This gene encodes a cysteine-rich (SPARC) multifunctional matrix-associated protein, which is associated with context- and cell-type dependent effects, ranging from tumor suppression to tumor cell invasion and metastasis [38]. The divergent activity of this protein was linked to the different functional activities of its proteolytic products [39]. The SPARC protein binds several extracellular matrix proteins and influences the secretion and activation of MMPSs. The elevated expression of SPARC was demonstrated to enhance chemosensitivity to drugs such as gemcitabine and 5-fluorouracil [40]. Furthermore, the downregulation of SPARC was observed in 5-fluorouracil and irinotecan-resistant colon cancer cells, and its re-expression was observed to restore sensitivity [41,42]. Furthermore, SPARC overexpressing tumors were reported to be highly sensitive to chemotherapy, with the N-terminal domain of SPARC being highly efficacious in enhancing apoptosis via blocking inhibitory interaction of Bcl with caspase 8 [39]. This result suggests that the downregulation of SPARC, as observed in our study, is a cellular strategy for inducing resistance and progression to more aggressive phenotypes in late-stage/metastatic melanomas. This context is yet to be fully explored and warrants investigation.

In addition to efforts directed at understanding the relationship between drug sensitivity and the genomic and genetic characteristics of cancer cell lines, major efforts have been invested in relating the findings from in vitro studies of cancer cell lines to in vivo trials.

## 4. Methods

### 4.1. Datasets

Five datasets from microarray or RNA sequencing of patient-derived biopsies (patients were treated with BRAF/MEK inhibitors) were collected from the Gene Expression Omnibus (GEO) datasets repository (https://www.ncbi.nlm.nih.gov/gds/ (accessed on 22 July 2022)). The accession numbers for these datasets are GSE99898 [43], GSE61992 [44], GSE50509 [45], GSE77940 [46], and GSE65185 [19]. A merged version of the five datasets, named “FiveDatabase” and comprising 197 samples, was achieved through computing the z-score of normalized counts for each common gene between the datasets [47]. Data from the project with GEO accession number GSE99898 was used as the primary dataset. It comprised biopsy samples obtained before treatment (17 samples—“PRE”) and after BRAF/MEK inhibitor treatment relapse (13 samples—“PROG”).

### 4.2. The 13-Risk Panel Establishment and 13-Risk Score Calculation

In our previous study, we identified 18 human proteins differentially expressed between vemurafenib-resistant A375 cells and compared to parental using a Human XL Oncology Array (Cat# ARY026; R&D Systems, Minneapolis, MN, USA) [22]. A total of 262,143 permutations/combinations between the 18 genes were tested to predict the resistance stage in GSE99898 samples, and validated in the other datasets and the “FiveDatabase”. Classifier markers were estimated after a receiver operative curve (ROC) analysis performed with the OptimalCutpoint R package (version 1.1-5). The area under the curve (AUC), positive predictive value (PPV), negative predictive value (NPV), sensitivity, and specificity were used as classifier metrics. The combination of genes with the highest AUC was selected.

A score, which was named the 13-gene risk score, was calculated according to the following equation: $\sum x_{i}*\beta_{i}$, with $x_{i}$= expression value of DEG i, and $\beta_{i}\;$= regression coefficient of gene i from a linear discriminant analysis based on pre-treatment (sensitive) or progression (resistant) stage.

A fold change (FC) of the 13-risk score between PROG and PRE samples was calculated for each sample and stratified as “Down” if FC < 0, “Slightly up” if 0 < FC < z, where z = 0.25 * max (FC), and “Up” if FC > z.

### 4.3. KEGG Pathways Enrichment Analyses

Pathway enrichment analyses were performed with the pathfinder R package (version 2.0.0) [48] using the active-subnetwork-oriented enrichment method. Differentially expressed genes were obtained through comparing PROG/resistant to PRE/pre-treatment samples. Genes with a *p*-value <0.1 were selected. Pathways were clustered following a principal component analysis that computed the agglomerate z-scores across the samples analyzed.

### 4.4. Protein-Protein Interaction Network

Protein–Protein interactions were collected using the NetworkAnalyst platform (https://www.networkanalyst.ca (accessed on 15 January 2023)). Only gene pairs with correlation coefficients >|0.3| were further selected, and a network was built using Cytoscape 3.8 [49].

### 4.5. Statistical Tests and Plots Design

Statistical tests were performed with R software, with Kruskal.test(), cor.test(), lm(), and lda() functions for Kruskal–Wallis test, Spearman’s correlation, linear regression analysis, and linear discriminant analysis, respectively.

Hierarchical clustering and plotting were performed with the pheatmap [50] R package (version 1.0.12), and the optimized number of clusters was estimated with the clValid R package (version 0.7). Principal component analyses (PCA) and PCA clustering was performed with FactoMineR (version 2.8) and plotted with factoextra (version 1.0.7) packages. Boxplots, Barplot, and Dotplots were designed with the ggplot2 [51] R package (version 3.4.2).

## 5. Conclusions

Our approach for predicting melanoma progression leverages both proteomics results and gene expression datasets derived from different populations to yield interesting mechanistic insights on adaptive resistance to BRAF/MEK inhibitors in melanomas. Selecting genes for proteins that are critical in several cancer pathways related to drug sensitivity, our classifier approach rigorously identified patterns that are suggestive of varying influence of genes on the progression of melanomas in gaining resistance to targeted inhibitor therapy. Derived from patient biopsies, these results are clinically relevant and can be interpreted to assist in the treatment and management of melanoma patients in the future. Moreover, based on differentially expressed genes that were predictive of treatment resistance, we proposed potential mechanisms for the development of BRAF/MEK inhibitor resistance, particularly the dysregulation of HMOX1 and SPARC. Lastly, our results allowed the postulation of a biologically informative model that is supported in large part based on evidence in the literature and opens areas requiring further probing. While the utilization of machine learning models and approaches in answering critical questions in cancer biology and clinical care is not without faults, continued research in this area will be beneficial in delivering personalized care that leverages the profiling of genes and proteins to optimize treatment protocols.

## Figures and Tables

**Figure 1 ijms-24-08407-f001:**
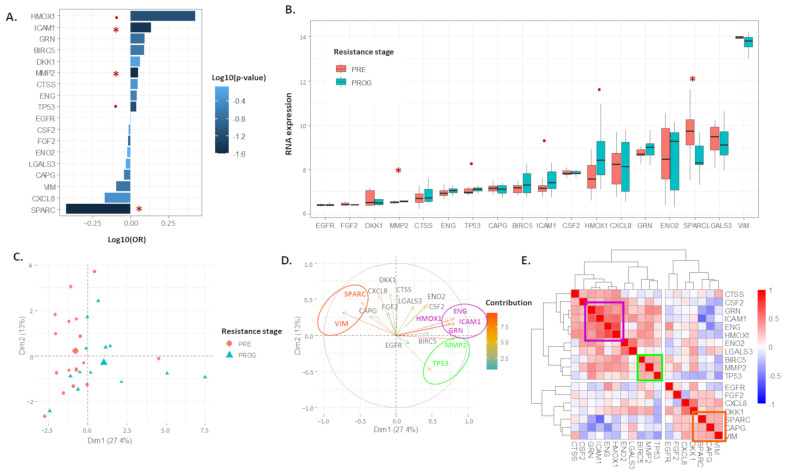
The 18 proteomic panel was deregulated in human melanoma resistant model (GSE99898). (**A**). Barplot representing odd ratio (OR) of a linear regression analysis of 18 proteomic genes according to resistance stage. (**B**). Boxplot representing expression distribution of 18 genes according to resistance stage. (**C**,**D**). Principal component analysis of 18 genes explaining heterogeneity observed across samples with individual/samples (**C**) and variables/genes (**D**) plots. Red asterisk (*) *p*-value < 0.05, red dot (.) *p*-value < 0.1. (**E**). Heatmap representing hierarchical clustering of Spearman’s correlation matrix of 18 proteomic genes.

**Figure 2 ijms-24-08407-f002:**
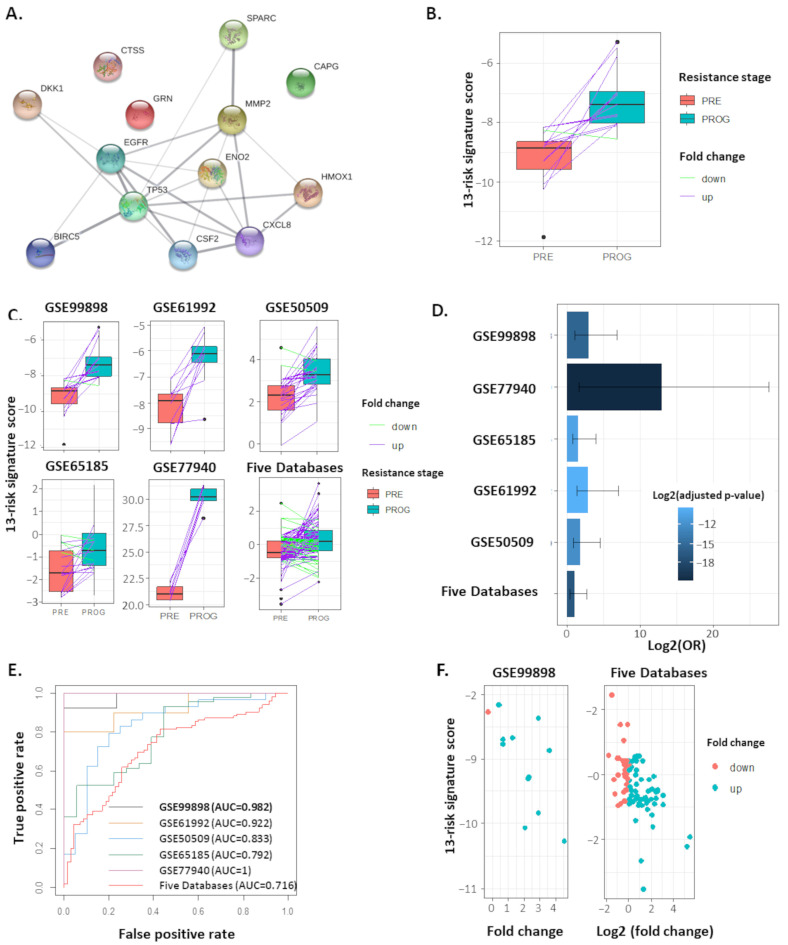
An optimized 13-risk panel predicts resistance stage. (**A**). Protein–protein interaction (PPI) and relationship network between 13 genes (network obtained from STRING database). (**B**,**C**). Boxplot representing value distribution of 13-risk score according to resistance stage in GSE99898 (**B**) and other databases (**C**), with paired samples fold change highlighted. (**D**). Barplot representing odd ratio (OR) of linear regression analysis of 13-risk score according to resistance stage. (**E**). Area under curves of Receiver operative curves analysis of 13-risk score to predict resistance stage in different databases analyzed. (**F**). Dotplots representing value of 13-risk score in PRE samples, differentiated through their score fold change.

**Figure 3 ijms-24-08407-f003:**
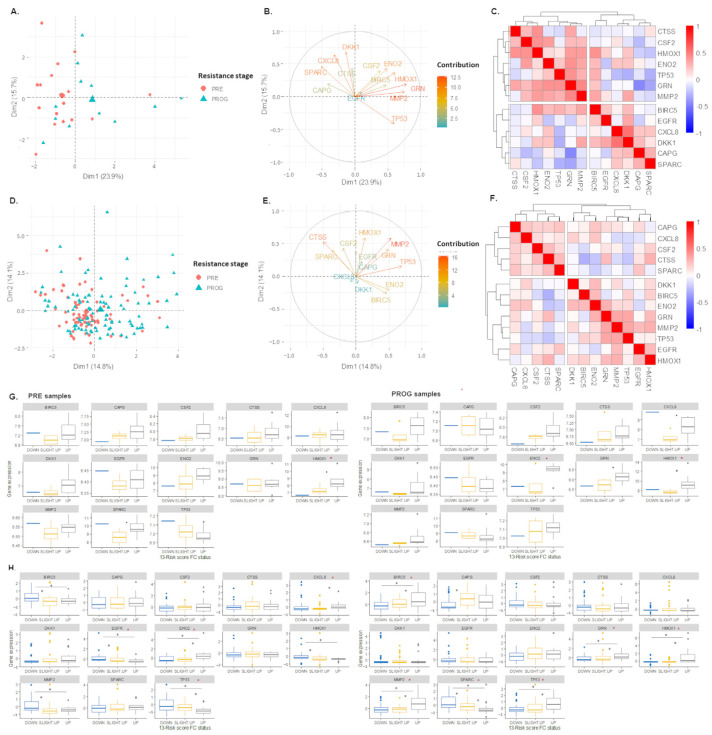
This study’s 13-risk score is highly differentiated according to resistance stage. (**A**,**B**,**D**,**E**). Principal component analysis of 13-risk panel explaining heterogeneity observed in GSE99898 (**A**,**B**) and FiveDatabases (**D**,**E**) samples, with individuals/samples (**A**,**D**) and variable/genes (**B**,**E**,**C**,**F**). Heatmap representing hierarchical clustering of Spearman’s correlation matrix of 13-risk genes panel observed in GSE99898 (**C**) and FiveDatabases (**F**) samples. (**G**,**H**) Boxplot representing distribution of 13-risk genes according to 13-risk score fold change, stratified as Down, Slightly up, and Up. Black dot (.), *p*-value < 0.1 for Up vs. Slightly up comparison; black asterisk (*), *p*-value < 0.05 for Up vs. Slightly up comparison; red dot (.), *p*-value < 0.1 for Up vs. Down/Slightly up comparison; and red asterisk (*), *p*-value < 0.05 for Up vs. Down/Slightly up comparison.

**Figure 4 ijms-24-08407-f004:**
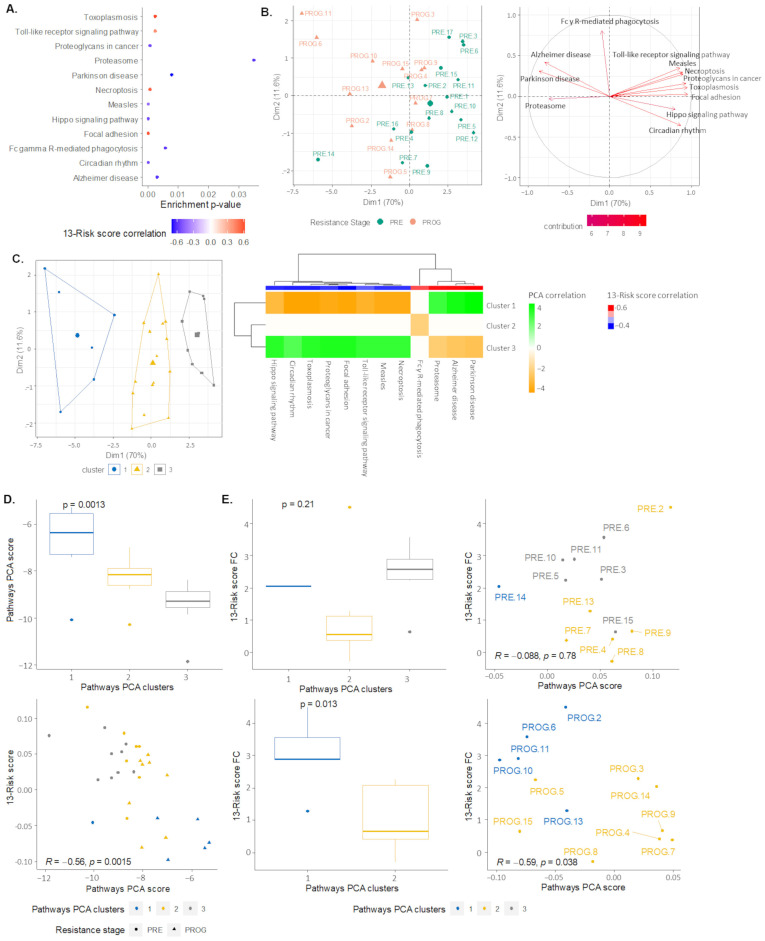
A subset of pathway signaling is 13-risk score-dependent to modulate therapeutic sensitivity of human melanoma GSE99898 samples. (**A**) Dotplots representing enrichment *p*-value of 12 pathways that present a similar significant correlation with 13-risk score in GSE99898 and FiveDatabases samples. (**B**) Principal component analysis (PCA) of 13 pathways’ scores (right) explaining variance observed in GSE99898 samples (left), differentiated through their resistance stage. (**C**) Hierarchical clustering of previous PCA identifying three clusters of samples (left) and pathway scores according to clusters samples (right). (**D**) A 13-risk score expression according to pathways PCA clusters (top) and score (bottom) in samples. (**E**) A 13-risk score fold change (FC) according to pathways PCA clusters (left) and score (right) in PRE (top) and PROG (bottom) samples. Kruskal–Wallis *p*-value (significant < 0.05) for boxplots, and Spearman’s correlation coefficient with corresponding *p*-value for dotplots.

**Figure 5 ijms-24-08407-f005:**
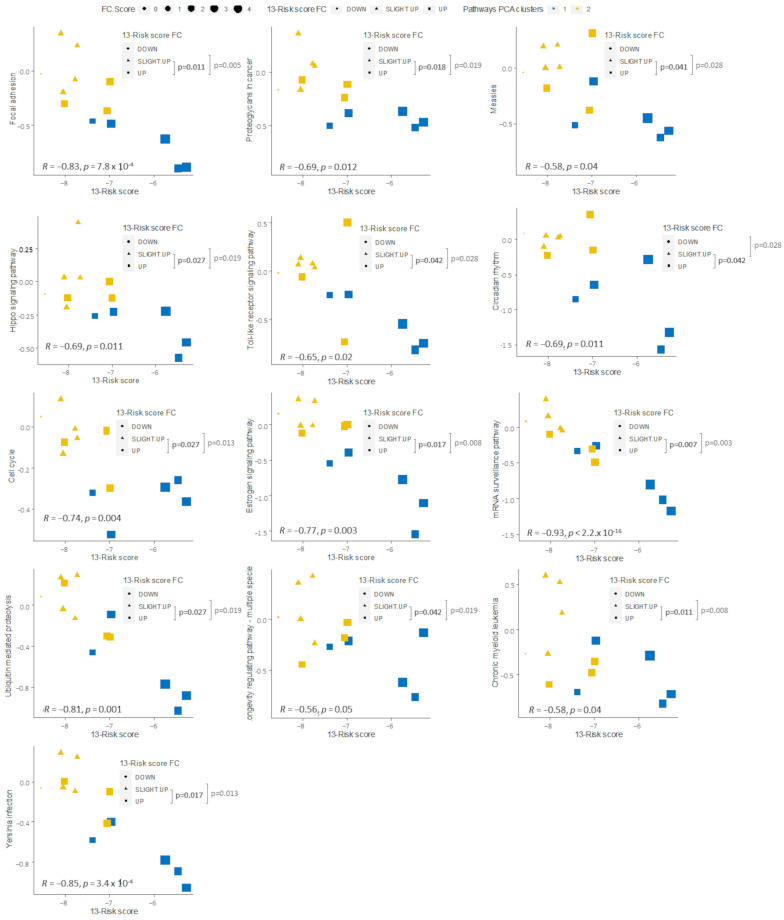
Pathways score distribution according to 13-Risk score and 13-risk score fold change (FC), stratified as downregulated, slightly upregulated, and upregulated in the PROG GSE99898 samples. Spearman’s correlation with corresponding *p*-values and Kruskal–Wallis tests for the 13-Risk score FC comparison (*p*-value significant < 0.05).

**Figure 6 ijms-24-08407-f006:**
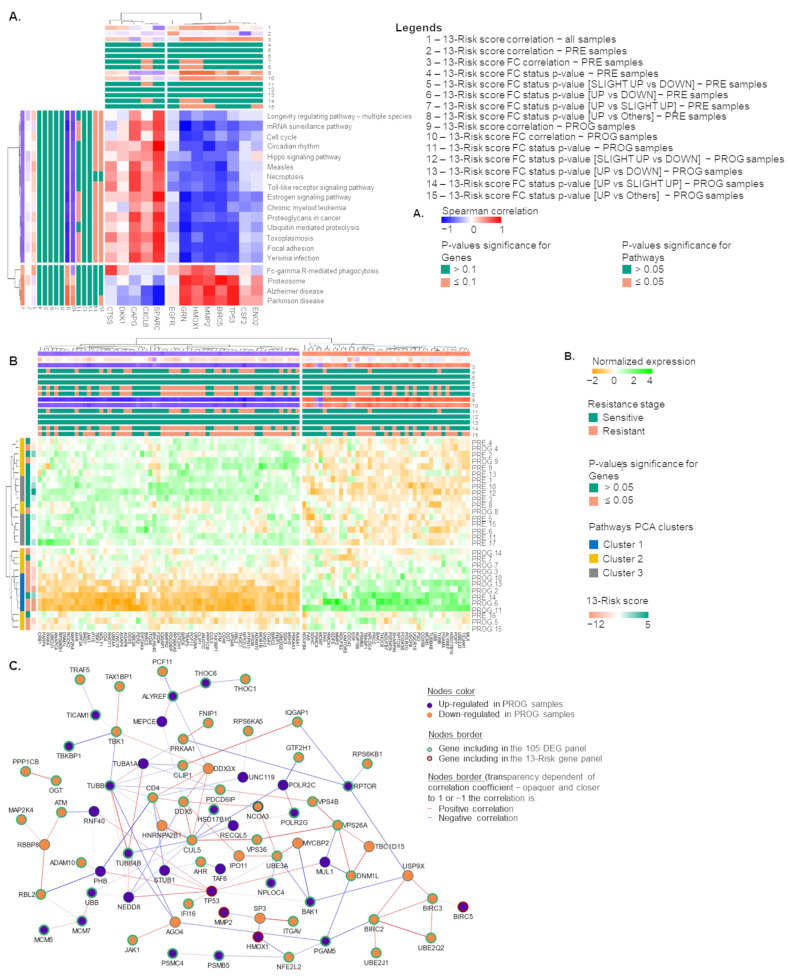
A total of 105 genes involved in selected pathways are dependent on 13-risk score to modulate therapeutic sensitivity of melanoma samples. (**A**). Hierarchical clustering of Spearman’s correlation matrix between 19 selected pathway scores and 13-risk genes panel. (**B**). Hierarchical clustering of normalized expression of 105 DEGs selected in GSE99898 samples. Annotations described correlation and Kruskal *p*-values for genes and pathways according to 13-risk score and 13-risk score fold change. (**C**). Protein interaction network obtained from NetworkAnalyst platform between 105 genes and 13-risk genes panels. Only correlation > |0.3| were collected.

**Figure 7 ijms-24-08407-f007:**
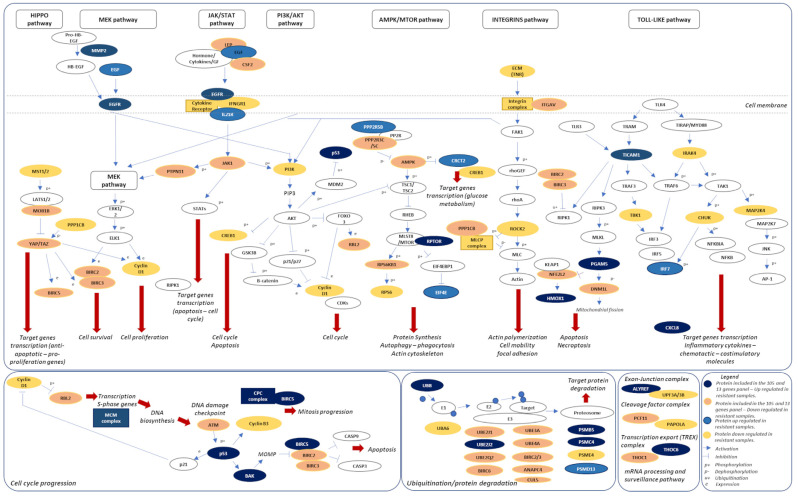
KEGG pathway model illustrating main genes involved in 13-risk score dependent BRAF inhibitor resistance process. GF, growth factor; CPC, chromosome passage protein complex; MCM, minichromosomal maintenance; MOMP, mitochondrial outer membrane permeabilization; and MLCP, myosin light chain phosphatase.

**Table 1 ijms-24-08407-t001:** A list of 18 genes selected from proteomic analysis of sensitive and resistant A375 melanoma cell line.

HO-1/HMOX1
Endoglin/CD105
CXCL8/IL8
CTSS
GRN
LGALS3
BIRC5
GM-CSF
ICAM-1
DKK-1
SPARC/BM-40
CAPG
P53
FGF2
ENO2
MMP2
EGFR
VIM

**Table 2 ijms-24-08407-t002:** Table summarizing three classifier markers obtained after a receiver operative curve (ROC) analysis to estimate resistant stage predictive capacity of each gene, positive predictive value (PPV), negative positive value (NPV), and area under curve (AUC) of ROC analysis.

	Classifier Markers		
	PPV	NPV	AUC
ENG	0.636	0.684	**0.633**
HMOX1	0.700	0.700	**0.706**
ICAM1	0.778	0.714	**0.697**
GRN	0.778	0.714	**0.674**
VIM	0.667	0.722	**0.624**
TP53	0.600	0.733	**0.683**
MMP2	0.750	0.778	**0.796**
SPARC	0.769	0.824	**0.76**
FGF2	0.520	1.000	**0.615**
BIRC5	0.750	0.682	**0.606**
CTSS	0.522	0.857	0.593
CAPG	0.667	0.667	0.588
CXCL8	0.800	0.640	0.579
DKK1	0.500	0.833	0.552
LGALS3	0.529	0.692	0.548
ENO2	0.625	0.636	0.511
CSF2	0.464	1.000	0.507
EGFR	0.464	1.000	0.498

**Table 3 ijms-24-08407-t003:** Table summarizing statistical analyses carried out to estimate capacity of 13-gene panel to predict resistance stage in five different datasets and merging of five datasets: a Kruskal–Wallis test, linear regression with odd ratio (OR) estimation, receiver operative curve (ROC) analysis with classifier markers estimation, positive predictive value (PPV), negative positive value (NPV), and area under curve (AUC) of the ROC analysis.

	Kruskal-Wallis Test		Linear Regression				Classifier Markers					
	*p*-Value	Adjusted *p*-Value	OR	95%CI OR	*p*-Value	Adjusted *p*-Value	Sensitivity	Specificity	PPV	NPV	AUC	95%CI AUC
GSE99898	8.30 × 10^−6^	4.26 × 10^−5^	7.38	3.58–15.19	8.70× 10^−6^	2.10 × 10^−5^	0.923	1.000	1.000	0.944	0.982	0.943–1.021
GSE61992	1.92 × 10^−3^	6.55 × 10^−3^	7.29	2.96–17.95	4.60 × 10^−4^	1.67 × 10^−3^	0.800	1.000	1.000	0.818	0.922	0.797–1.047
GSE50509	8.64 × 10^−5^	2.53 × 10^−4^	3.53	2.00–6.25	7.43 × 10^−5^	2.34 × 10^−4^	0.793	0.800	0.852	0.727	0.833	0.709–0.957
GSE65185	3.40 × 10^−4^	1.07 × 10^−3^	2.94	1.70–5.09	2.80 × 10^−4^	8.21 × 10^−4^	0.932	0.556	0.837	0.769	0.792	0.67–0.913
GSE77940	3.95 × 10^−3^	2.05 × 10^−2^	8163.08	2632–25310	2.40 × 10^−8^	4.91 × 10^−7^	1.000	1.000	1.000	1.000	1	1−1
FiveDataBase	1.47 × 10^−6^	3.76 × 10^−6^	2.11	1.56–2.86	3.37	8.63 × 10^−6^	0.814	0.571	0.735	0.678	0.716	0.639–0.794

## Data Availability

The datasets used for the study can be retrieved on https://www.ncbi.nlm.nih.gov/gds/, accessed on 3 April 2023.

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
