# Peer review of "A Predictive Model of Adaptive Resistance to BRAF/MEK Inhibitors in Melanoma"

_ijms, 2023, doi:10.3390/ijms24098407_

Round 1

Reviewer 1 Report

The authors evaluated the predictive capacity of a gene panel corresponding to proteins differentially abundant between treatment sensitive versus resistant cell lines to decipher predictors of treatment resistance and potential resistance mechanisms to BRAF/MEK inhibitor therapy in patient biopsy samples. The results highlighted the utility of some genes including HMOX1, ICAM, MMP2, and SPARC in predicting treatment outcomes and the underlying mechanisms that can be targeted to reduce the development of resistance to BRAF/MEK inhibitors. The results are clinically relevant and may be help to the treatment and management of melanoma patients. 

Due to tumor heterogeneity, cancer cells will inevitably undergo adaptive resistance after treatment with any targeted drug. Exploring schemes to identify heterogeneous molecules may be the most critical strategy and challenge from the desregulated genes and proteins and their associated signaling pathways.

In order to further enhance the scientific and clinical value of the research, it is suggested: (1) List the driven heterogeneous molecules identified from the database; (2) Scoring the driven heterogeneous molecules.

Author Response

Response to Reviewers

We sincerely thank the reviewers for their constructive observations, comments, and recommendations. We have gone over the manuscript and diligently made the recommended changes.

Moreover, we have noticed that there are still two points about the layout format that need to be revised before we can proceed with the review process.

- Firstly, we kindly request that you provide the table in the manuscript as an editable table, rather than as an image. Please note that the use of images for tables is not allowed in our journal.

Our response: We included three editable tables to the manuscript.

- Secondly, we request that you change the format of the references to a numbered format. Please follow our guidelines for reference formatting, which can be found on our website.

Our response: The references format was changed.

Reviewer 1:

To further enhance the scientific and clinical value of the research, it is suggested: (1) List the driven heterogeneous molecules identified from the database; (2) Scoring the driven heterogeneous molecules.

Our response: Besides being highly predictive of a BRAF/MEK inhibitor-resistant state, we identified the pathways involving functionally heterogeneous genes HMOX1 (and its regulators PGAM5, NFEL2L2, and KEAP1-E3 ligase) and SPARC mediators of adaptive resistance. Furthermore, we compared the transcriptomic landscape between drug-sensitive tumors and tumors with reduced sensitivity but not fully resistant/progressed, to identify other heterogenous candidate genes with expression progressively altered as resistance emerges and are critical roles in the transition of tumor cells to adaptive resistance. Examples of genes identified include PRC1 (a regulator of cytokinesis), CCNB2 (a cell cycle regulator), ACSL3 (an enzyme involved in fatty acid synthesis). The full result of this analysis is attached for your kind review (File name – DEGs – Early Resistance Stage). Overall, our results highly suggest that functionally different genes play a cooperative role in the transition to an insensitive state.

Classifier scores for each gene in our panel are provided in Table 2. It provides quantitative metrics on the ability of each individual gene derived from our proteomic panel to discriminate between treatment-sensitive and resistant stages and provides an estimation of the weight of each gene in the overall predictive capacity of the gene panel.

Reviewer 2 Report

This manuscript by Ruiz et al. utilized a previously identified proteomics panel against available datasets in order to generate a predictive model for resistant to BRAF/MEK inhibitors in melanoma. The authors identified a 13-gene signature panel that correlated with resistance to these treatments. The authors also provided mechanistic insights into the potential pathways involved in response to BRAF/MEK inhibitors.

Minor revisions:

-How many samples were included in the datasets used? What was the staging/grading of the tumors? What is the distribution of the patients regarding age, ethnicity, gender etc? It would be useful to include a table with this information.

-The p values should be better presented: 3.57*10-6 should be replaced by 3.57x10-6 

There are some grammatical errors and 

Author Response

Reviewer 2:

Reviewer: -How many samples were included in the datasets used? What was the staging/grading of the tumors? What is the distribution of the patients regarding age, ethnicity, gender etc.? It would be useful to include a table with this information.

The p values should be better presented: 3.57*10-6 should be replaced by 3.57x10-6 

Our action: Four out of the five datasets used in our analysis have data available on important clinical and genetic characteristics of the patients from whom biopsy samples were obtained. We have included these as an additional file to our submission. However, since this is already published in the cited publications from which these datasets were obtained through the Gene Expression Omnibus, we would recommend against publishing this as part of our paper.

Our action: The format in which the p-values were reported have now being changed as recommended by the reviewer and all grammatical errors corrected.

Reviewer 3 Report

The authors used machine learning involving protein expression datasets from BRAF/MEK inhibitor-sensitive and inhibitor-resistant melanoma patient biopsy samples to correlate likely changes in gene expression between the two groups. They subsequently implicated HMOX1 and SPARC expression as probable modulated pathways that may, at least partially, define cellular mechanisms associated with adaptive BRAF/MEK inhibitor resistance. This latter proposition was perhaps better supported by descriptions of associated published findings than by definitive outcomes determined through their modeled clusters.

Overall, I found this to be a meaningful analysis that, if accurate, could positively guide research emphasis toward more effective protein or RNA interference targets or prompt the study of potentially reduced resistance combination therapies.

Portions of the manuscript are poorly edit and should be revised prior to publication. Key examples include:

·         [line 42] “results” should be deleted

·         [line 251] “slightly up upregulated” should be corrected

·         [line 259] presumably, “it” should read ‘out’

·         Line [341] ‘of’ is missing before “BRAF/MEK”

·         Lines 391-392 are nonsensical as written

·         Line 409 must be rephrased for clarity of message

·         Figure 7 is a key graphic, but it is distorted and blurred. Was this taken from another published source? Perhaps higher resolution artwork of appropriate dimensions should be used instead.

Author Response

Reviewer 3:

Portions of the manuscript are poorly edit and should be revised prior to publication.

Figure 7 is a key graphic, but it is distorted and blurred. Was this taken from another published source? Perhaps higher resolution artwork of appropriate dimensions should be used instead.

Our response: We thank the reviewer for the keen eye in noticing these mistakes. They have now been corrected.

  •  [line 42] “results” should be deleted removed: now line 37 (‘results’ removed)
  • [line 251] “slightly up upregulated” should be corrected: now line 219 (extra "up" has been deleted)
  • [line 259] presumably, “it” should read ‘out’: now line 216 (‘it’ replaced with ‘out’)
  • Line [341] ‘of’ is missing before “BRAF/MEK”: now line 344 (the article ‘of’ now added)
  • Lines 391-392 are nonsensical as written: now line 393 (edited for clarity)
  • Line 409 must be rephrased for clarity of message: now line 396 (edited for clarity)

Furthermore, Figure 7 is an original (unpublished) figure. We have updated that figure to improve its resolution and included a separate high-quality .png file for it as part of our submission.